behaviour/cognition/psychology

PTS, HTQ, war, emotion, face

**Author for correspondence:**
Gustaf Gredebäck
e-mail: gustaf.gredeback@psyk.uu.se

# Social cognition in refugee children: an experimental cross-sectional study of emotional processing with Syrian families in Turkish communities

Gustaf Gredebäck[1], Sara Haas[1], Jonathan Hall[2],
Seth Pollak[3], Dogukan Cansin Karakus[4] and
Marcus Lindskog[1]

[1]Department of Psychology, and [2]Department of Peace and Conflict Research, Uppsala University, Uppsala, Sweden
[3]Department of Psychology, University of Wisconsin-Madison, Madison, Wisconsin, USA
[4]Göttingen Graduate School of Social Sciences, University of Göettingen, Göttingen, Germany

GG, 0000-0003-3046-0043; ML, 0000-0003-1326-6177

More than 5.6 million people have fled Syria since 2011, about half of them children. These children grow up with parents that often suffer from war-related mental health problems. In this study, we assess emotional processing abilities of 6–18 year-old children growing up in families that have fled from Syria and reside in Turkish communities (100 families, 394 individuals). We demonstrate that mothers', but not fathers', post-traumatic stress (PTS) impacts children's emotional processing abilities. A 4% reduction of mothers' PTS was equivalent to 1 year of development in children, even when controlling for parents' traumatic experiences. Making a small investment in increased mental health of refugee mothers might have a positive impact on the lives of their children.

## 1. Introduction

The civil war in Syria is the most severe armed conflict in the world during last three decades [1] with more than 5.6 million people having fled the country since its onset in 2011. In the city of Aleppo alone, fighting has continued for almost 5 years, resulting in more than 24 000 civilian casualties, and 'war crimes of historic proportions' [1, p. 2]. As a consequence of their

experiences, many refugees struggle with war-related mental health problems [2], with a prevalence of post-traumatic stress disorder (PTSD) at 11–14% among adults relocated to Germany [3,4] and above 30% for those living in refugee camps in Syria and neighbouring countries [5].

The extant literature suggests that such problems in parents are likely to be associated with difficulties in children [6]. Cross-sectional questionnaire studies from refugee camps with Syrian refugees demonstrate that children are at risk of developing behavioural problems related to poor emotional regulation [7], conduct problems [8] and poor mental health [9], especially if parents suffer from war-related mental health problems. Qualitative studies with Syrian refugees in Turkey indicate that parents face substantial difficulties tending to the needs of their children, including a shortage of resources, perceived chaos in their daily lives and lack of safety [10]. Parents also observe negative changes in children's behaviour and tend to adopt negative and harsh parenting styles to meet these changes [10]. Stressful living conditions, coupled with parents' suffering from poor mental health, has a negative impact on the quality of parent–child interactions [6,11]. This, in turn, might impact the social cognitive development of children [12]. However, to our knowledge this possibility has never been assessed in refugee populations.

Social cognitive development starts during infancy and continues throughout childhood, with long-term consequences across the lifespan [13–15]. For example, social cognitive ability has been identified as important as intelligence for labour-market success [16] as well as having large implications for the quality of social interactions, peer-relations and school performance [17]. One important dimension of social cognition is emotional processing, the ability to make reasonable inferences based upon other people's facial expression [18]. Studies with non-refugee Western children demonstrate that this ability is highly experience dependent and that our social encounters during childhood impact the way we attend to, and process, facial expressions later in life. For example, traumatic experiences early in life have a detrimental effect on emotional processing, with evidence for both specific (e.g. exposure to anger and violence leads to a heightened sensitivity to angry faces) and general (e.g. poor attachment quality leads to a broad deficit in emotional processing) effects [12,18].

The goal of this paper is to better understand the relationship between parents' experience of war, mothers' and fathers' subsequent post-traumatic stress (PTS), and children' social cognitive development, here operationalized as emotional processing [18]. We hypothesize that children whose parents experienced war-related traumatic events and as a consequence thereof suffer from poor mental health develop worse emotional processing than children whose parents that were less effected by the same events. It is, however, unclear from the prior literature if this hypothesized effect can be related to parents' experiences or their current mental health. To accomplish this goal, we combine cross-sectional experimental tasks (where each individual is given a series of psychological tests) and questionnaire data from 100 Syrian refugee families (394 individuals) living in Turkish communities. We measure children's (and parents') emotional processing and relate it to mothers' and fathers' self-reports of war-related PTS, and potentially traumatic experiences.

Most previous studies targeting refugee populations focus on adults and families living in Europe or in camps located in close proximity to the conflict [19]. However, it is unclear if findings from these groups generalize to refugees living in other contexts [19]. When it comes to refugees from Syria, the largest recipient country is Turkey, with more than 3.6 million registered refugees, most of whom live outside refugee camps, in communities in southern Turkey, close to the boarder to Syria [20]. With this paper we aim to shed a light on families that have not migrated to the European Union but live in communities outside refugee camps in a country neighbouring that of their origin. This group, representing the majority of refugees from Syria have, to date, remained outside the focus of the scientific literature. For existing work with this group of refugees see [8,10,11].

## 2. Method

### 2.1. Participants

One hundred Syrian families living in Konya, Turkey participated in the study. The sample consisted of 174 adults ($M_{age} = 39.8$, s.d.$_{age} = 7.8$ range = (22, 60), 55.7% women) and 220 children ($M_{age} = 12.2$, s.d.$_{age} = 3.1$, range = (6, 18) 42.3% girls). The median number of children in each family was 2, range = (1, 5). The vast majority of families originate from Aleppo ($n = 151$), with a small group of families from Ar Raqqah ($n = 1$), Damascus ($n = 4$), Deir al-Zour ($n = 2$), Homs ($n = 1$), Idlib ($n = 4$) and Lattakia ($n = 6$).

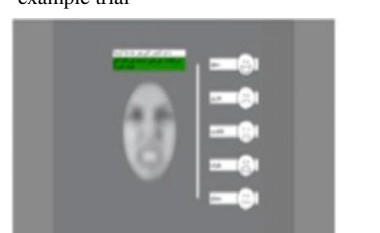 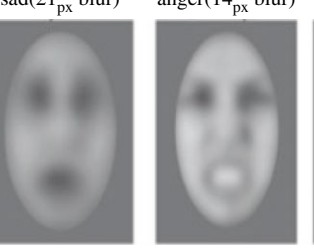 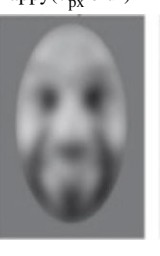 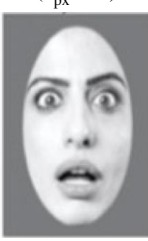

example trial sad(21$_{px}$ blur) anger(14$_{px}$ blur) happy(7$_{px}$ blur) fear(0$_{px}$ blur)

**Figure 1.** Stimuli used in the emotional processing task assessing social cognition. The text on the example trial can be translated as 'How does this person feel? From the words on the right, click which one you think is best'.

## 2.2. Procedure and design

Research assistants (fluent in Arabic and Turkish), supervised by co-author D.C.K., visited the homes of these refugee families between October 2019 and January 2020. Recruitment was based on an opportunistic sampling procedure where participating families were asked to recommend the study to other families, as records of refugee families with children in our target community setting was not available. The initial family was referred from the personal network of D.C.K. The study protocol lasted approximately 60 min for adults over 18 years and 30 min for children. Multiple family members participated in parallel. Sometimes children needed help (instructions) from parents, which made the length of the session highly variable. The study was approved by the regional ethics review board in Sweden (2018-395) and the Necmettín Erbakan Universitesi in Turkey (2019/17). Each family received a monetary compensation equivalent of 10 Euro per participant for participation.

The study started with tea and biscuits (brought by the research team), and written and verbal informed consent from all participants. Following this, each family member sat in front of a computer (DELL Vostro 3568, 15′ screen), with noise-cancelling headphones while completing a larger battery of experimental tasks assessing IQ [21], visual attention [22], emotional processing [23], working memory [24] and risk-taking [13] with written instructions in Arabic. Adult participants also completed a series of questionnaires (all in Arabic) assessing demographics, social environment, migration history, risk factors, discrimination, potentially traumatic events and PTS [25–28] after the experimental tasks (see the electronic supplementary material for complete list). The current paper reports on emotional processing of children as well as potentially traumatic events and PTS of mothers and fathers (parents' emotional processing is also analysed to control for overall social cognition of the family). The other tasks will be analysed and reported in separate papers, none of these additional experimental tasks or questionnaires have been analysed or related to emotional processing in preparation of this paper. To minimize the strain and psychological distress of vulnerable participants in this context, children only performed the experimental tasks. Parents helped children understand the tasks if needed. Two researchers (one male, one female) supervised each session.

The emotional processing task included photos of eight adult (four female), Arab faces expressing anger, happiness, sadness, fearful and a neutral expression. To make the task more difficult, and simulate the difficulty of everyday emotional processing, each image was altered using four Gaussian blur levels (clarity levels: 21px, 14px, 7px, 0px) to create 160, 293 × 443 pixel ellipse images, with a total display area of approximately 5% of the screen. The centre of each image was located at 15.43° × 10.98° visual degrees (768px by 540px) and the image extended from 12.56 to 18.19° (X) by 6.53–15.3° (Y) visual degrees (figure 1). Presentation was blocked such that the most difficult clarity level came first (21px blur) followed by 14px, 7px and 0px, respectively. Image presentation was randomized within blocks. Each trial included a fixation cross (1500 ms) followed by an image with response options asking whether the face expressed anger, happiness, sadness or fear, using text labels (5000 ms) appearing on the right side of the image. Each image was rated by 37 university students at the Necmettín Erbakan Universitesi to confirm that the photos were easily categorized and discriminated (see the electronic supplementary material for rating descriptives). Initial pilots from Uppsala and Konya with refugee children of the target ages understood the task and responded correctly. The dependent variable is described in the analysis section below.

The PTS questionnaire, PTSD CheckList-Civilian Version (PCL-C$_{Abbreviated}$) [25], included six questions related to disturbing memories, negative emotions related to past experiences, avoidance of situations that remind of stressful experiences, distance to other people, irritation and anger and

concentration difficulties. For each item, responses were provided on a 5-point scale (from *not at all* to *extremely*) asking how much they have been bothered by each problem in the last month. The predictor variable used in the analysis was the aggregate score ranging from 5 (not been bothered by any of these problems during the last month) to 30 points (extremely bothered by all problems during the last month).

The Harvard trauma questionnaire (HTQ) [26] included a list of 16 war-related potentially traumatic events with tick boxes for each, asking if they experienced each in their lives prior to arriving in Turkey. Items were presented in a fixed order, as listed in figure 1. The predictor variable used in the analysis was the number of different events that participants had experienced.

## 2.3. Analysis

We investigated the effect of PTS and experiencing traumatic events (HTQ) on emotion discrimination using logistic generalized mixed models (LGMM) with a logit link function. Note that using the LGMM is equivalent to modelling a psychometric function using for example sensitivity measures such as d-prime with the added benefit of the possibility to model trial level data and evaluating the effect of multiple factors simultaneously on sensitivity [29].

For both children and adults, we evaluated two separate LGMM [29]. The first model evaluated the effects of PTS and HTQ on overall emotional processing together with the effects of age and sex. The second model included, in addition to any significant predictors from models $1_A$ and $1_C$, emotion and clarity level as predictors. With the second model, we aimed to investigate if any potential effects of PTS and HTQ on emotional processing also held while considering that the task contained different emotions presented at different clarity levels. All models were run on trial level data with response (correct = 1/wrong = 0) as the dependent variable. The models on child data had PTS and HTQ of their mothers and fathers, respectively, sex and age (model $1_C$) as well as emotion and clarity level (model $2_C$) as fixed effects and participant and family unit as random effects. The models on adult data included PTS, HTQ, sex (male = 0, female = 1), age (model $1_A$) as well as emotion and clarity level (model $2_A$) as fixed effects and participant as a random effect. As a follow-up analysis separate models were run for each emotion assessing the specific effect of PTS and HTQ on the five emotional expressions used in the emotional processing task, using the same procedure as outlined for model $1_C$.

The data used to run the above mentioned analysis is available at OSF (https://osf.io/qjrav) [30].

# 3. Results

## 3.1. Parent's mental health

The mental health, and the number and quality of potentially traumatic events, of parents was highly variable. PTS scores averaged 17.6 (s.d. = 5.5, range = (6, 30), with a cut-off for PTSD at 14 points) and the mean exposure to potentially traumatic events averaging 7.3 (s.d. = 2.8, range = [0, 15]). Based on this cut-off, it is estimated that 81% of mothers and 71% of fathers in our sample suffer from PTSD (based on thresholds for PCL-C$_{Abbreviated}$). PTS scores and the number of potentially traumatic events experienced correlated positively for both mothers ($r = 0.24$, $p = 0.002$) and fathers ($r = 0.39$, $p = 0.00045$). Experiencing more potentially traumatic events was associated with worse mental health. For details of mothers' and fathers' experiences see figure 2.

## 3.2. Children's emotional processing

Overall, children performed the emotional processing task well. On average, children identified 62% of emotions correctly. They found it easiest to identify happy emotions (84.4%), followed by neutral (69.2%), anger (58.4%), fear (51.3%) and sad (48.1%) expressions. Identification also improved with increased clarity, from 38.4% at maximum blur to 81.4% at full clarity. We investigated the effect of parents' PTS and experiences of potentially traumatic events on children's emotion discrimination. As noted above, the LGMM for modelling these effects included mothers' and fathers PTS and HTQ as separate fixed effects. Model $1_C$ included, in addition to parents' PTS and HTQ, children's age and sex. The results of model $1_C$ for the child data are summarized in table 1. As is evident from the table, there were two significant fixed effects, mothers' PTS and age. Children became better at discriminating emotions with age ($\exp(B) = 1.05$, 95% confidence interval (CI) = (1.024, 1.068)). That is, for each year the child

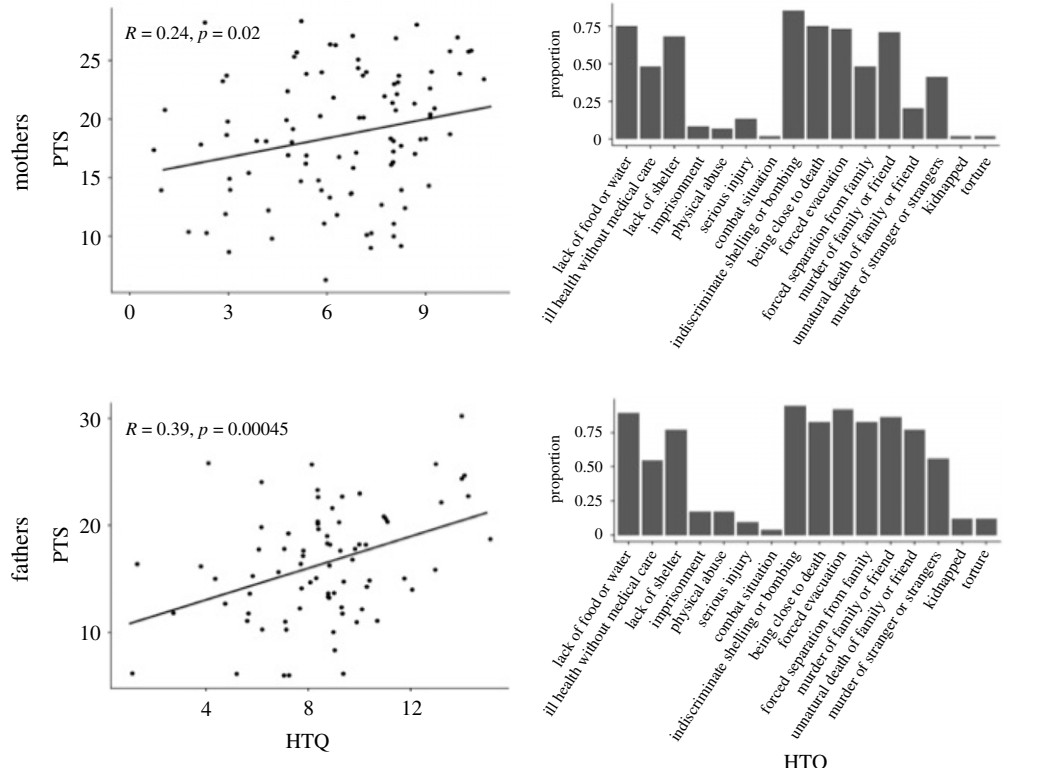

**Figure 2.** Descriptive data on potentially traumatic events and post-traumatic stress in mothers and fathers (along with correlation between these data) in the sample.

**Table 1.** Fixed effects parameter estimates for model 1$_C$ on adult and model 1$_A$ children data. (HTQ refers to the scale used to assess potentially traumatic events, PTS to the post-traumatic stress scale. Significant effects ($p < 0.05$) are marked in italics.)

| effect | B | s.e. | exp(B) | 95% exp(B) CI low | high | z | p-value |
|---|---|---|---|---|---|---|---|
| model 1—children | | | | | | | |
| intercept | 0.15 | 0.24 | 1.16 | 0.72 | 1.87 | 0.62 | 0.53 |
| HTQ—father | 0.02 | 0.02 | 1.02 | 0.99 | 1.06 | 1.31 | 0.19 |
| HTQ—mother | 0.02 | 0.02 | 1.02 | 0.97 | 1.06 | 0.73 | 0.47 |
| PTS—father | 0.00 | 0.01 | 1.00 | 0.99 | 1.02 | 0.52 | 0.61 |
| PTS—mother | −0.03 | 0.01 | 0.97 | 0.96 | 0.99 | −3.68 | *<0.001* |
| age | 0.05 | 0.01 | 1.05 | 1.02 | 1.07 | 4.18 | *<0.001* |
| sex (m–f) | 0.02 | 0.07 | 1.02 | 0.90 | 1.16 | 0.28 | 0.78 |
| model 1—adults | | | | | | | |
| intercept | 0.42 | 0.08 | 1.51 | 1.31 | 1.76 | 5.52 | <0.001 |
| HTQ | 0.02 | 0.02 | 1.02 | 0.99 | 1.06 | 1.43 | 0.15 |
| PTS | −0.01 | 0.01 | 0.99 | 0.98 | 1.01 | −1.11 | 0.27 |
| age | 0.00 | 0.01 | 1.00 | 0.99 | 1.01 | −0.37 | 0.71 |
| sex (m–f) | *0.31* | *0.11* | *1.37* | *1.11* | *1.69* | *2.89* | *<0.05* |

became older, the odds of correctly identifying an emotion became 1.05 times larger. Perhaps most importantly, children of mothers with higher PTS performed worse than those with lower PTS (exp(B) = 0.97, 95% CI = (0.96, 0.99)). Increasing mothers' PTS score with one point resulted in the

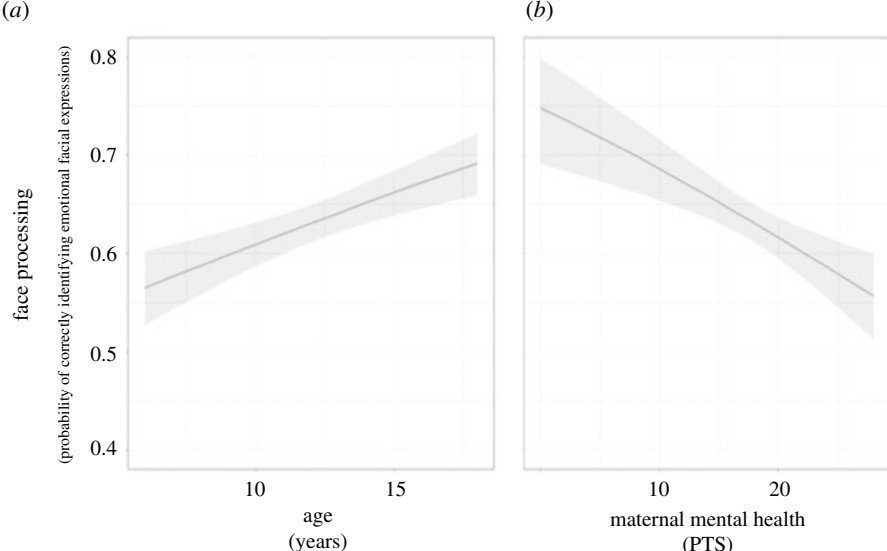

**Figure 3.** The marginal effect of age in years (*a*) and mother's mental health, PTS (*b*) on the probability of correctly identifying emotions in refugee children. Grey band indicates 95% confidence interval.

odds of correctly identifying emotions becoming 0.97 times lower. For an illustration of both effects see figure 3. In statistical terms, both these effects are considered small [31].

We ran a second model including the significant predictors from model $1_C$ together with emotion and clarity level as fixed factors (see the electronic supplementary material for full table of results). Importantly, while controlling for both emotion and clarity, the effects of mothers' PTS and age remained. Thus, children to mothers' that reported high PTS performed worse at emotional processing even when taking into consideration that different emotions and different clarity levels were used in the emotional processing task. We further evaluated the effects of age and mother's PTS on emotional processing by running separate models for each of the five emotions. The results are summarized in table 2. As is evident from the table, the effect of mother's PTS is significant for the sad and happy emotions and marginally significant for the neutral emotion. Although it is not significant in the anger and fear emotions, the direction of the parameter estimate is the same as for the other emotions.

In order to better understand the connection between parents' war experiences and children's emotional processing we repeated the same analysis for the parents' emotional processing and related this to their own self-reported war experiences (PTS and potentially traumatic experiences). The specific goal of this analysis was to see if the association between mothers' PTS and children's emotional processing is specific to children or if mothers well-being impacts her own social cognitive abilities, an association that might have a direct impact on children's social cognitive skill.

## 3.3. Parents' emotional processing

Parents correctly identified emotions in 63.8% of all presentations. It was easiest to identify happy emotions (88.1%), followed by neutral (65.0%), anger (62.6%), fear (54.2%) and sad (49.0%) expressions. Mothers identified 66% of the emotions correctly, with a higher rate of recognition for happy (89.1%) than sad (52.0) and increased performance with increases in emotional clarity (from 43.0 to 80.0% over clarity level). Similar results for fathers were obtained for happy (86.7%) and sad (44.7) with an increased performance following increases in emotional clarity (from 38.5 to 77.3% over clarity level). As is evident from table 1, neither parents' potentially traumatic experiences nor their current mental health impacted their emotional processing. The only significant effect was that of sex, with mothers outperforming fathers ($\exp(B) = 1.37$, 95% CI = (1.11, 1.69)). Thus, the odds of a mother correctly identifying an emotion were 1.37 times larger than that of a father. For more information about model $2_A$ that also controls for emotion and clarity, see the electronic supplementary material.

## 4. Discussion

The parents participating in the study have lived through much hardship. The majority had experienced living in a war zone, including indiscriminate shelling and bombing, forced evacuation and separation

**Table 2.** Fixed effects parameter estimates of mother's PTS and children's age for the separate emotions. (PTS refers to the post-traumatic stress scale. Significant effects ($p < 0.05$) are marked in italics.)

| effect | B | s.e. | exp(B) | 95% exp(B) CI low | high | z | p-value |
|---|---|---|---|---|---|---|---|
| **sad** | | | | | | | |
| PTS—mother | −0.03 | 0.01 | 0.97 | 0.96 | 0.99 | −3.14 | *0.002* |
| age | 0.04 | 0.01 | 1.05 | 1.02 | 1.07 | 3.25 | *0.001* |
| **anger** | | | | | | | |
| PTS—mother | −0.01 | 0.01 | 0.99 | 0.97 | 1.01 | −0.81 | 0.418 |
| age | 0.05 | 0.01 | 1.05 | 1.02 | 1.07 | 3.68 | *<0.001* |
| **happy** | | | | | | | |
| PTS—mother | −0.03 | 0.01 | 0.97 | 0.94 | 1.00 | −1.99 | *0.047* |
| age | 0.09 | 0.02 | 1.09 | 1.05 | 1.14 | 4.15 | *<0.001* |
| **neutral** | | | | | | | |
| PTS—mother | −0.02 | 0.01 | 0.98 | 0.95 | 1.00 | −1.94 | 0.052 |
| age | 0.04 | 0.02 | 1.04 | 1.01 | 1.07 | 2.64 | 0.008 |
| **fear** | | | | | | | |
| PTS—mother | −0.02 | 0.01 | 0.98 | 0.97 | 1.00 | −1.73 | 0.084 |
| age | 0.06 | 0.01 | 1.06 | 1.03 | 1.09 | 4.00 | *<0.001* |

from family, lack of shelter, food and water. More severe consequences of war, such as being close to death and witnessing the murder of family and/or friends, are also frequent (figure 2). Against this background, it is perhaps unsurprising that parents' mental health is poor.

Mothers who suffer from many PTSD related symptoms risk raising children with poor emotional processing. The effect is broad and impacts a wide range of emotional expressions. It does not appear to be centred on a single expression, as sometimes reported with respect to physical violence or neglect in Western contexts [18]. Most striking is perhaps that a small reduction in the PTS symptoms of mothers (by one point of the PTSD scale with a range from 6 to 30 points) is, statistically speaking, equivalent to 1 year of development in children (figure 3). It should be noted that although the effect of age and PTSD were stable and clear, they were also, in statistical terms, small effects. However, even small statistical effects can have important practical implications. Thus, we note that a small alleviation in PTS for mothers (approx. 4% change in PTS) can have an impact on children's long-term development (equivalent to 1 year of development on emotional processing). As outlined in the introduction, such problems are likely to have long-term consequences for these children throughout life [13–17]. The mechanisms that give rise to the association between maternal mental health and children's emotional processing is beyond the scope of the current paper but increased hostility and conflict within the family [32], negative and harsh parenting styles [10], parental withdrawal, emotional contagion and/or poor parental regulation of negative emotions [33] are potential candidates.

At the same time, fathers' experiences and mental health do not impact children's development in this sample. This finding stands in stark contrast to previous studies in other contexts where fathers' war experiences and mental health have been related to family outcomes and child experiences [34]. It is beyond the reach of the current study to firmly conclude why this is the case, but factors related to family structure, gender roles, gender reporting differences and a need for fathers to seek work outside the home might all contribute [10,11]. As illustrated in figure 2, mothers' and fathers' experiences are quite similar and a clear relationship between prior experiences and current mental health exist for both groups. Accordingly, the likelihood of us observing an association between child development and fathers' mental health is most likely not lower than that for mothers', which further strengthens the notion that it is the mothers who influence child development to a much larger degree than fathers.

In this context, it is important to note that mothers' and fathers' own experiences and PTS did not impact their own emotional processing. The ability to make inferences based upon other people's facial expression is an ability that develops early in life during childhood [35] and most likely the neural networks and experiences needed to process this information are already cemented in place, untouched by hardship of refuge and war experienced as adults. In addition, the association between maternal mental health and child development is present even when controlling for the potentially traumatic events experienced by both parents. This suggests that it is not the joint traumatic experience of the family and the child or the current living context of the family, which is the cause of the observed effect. Based on these findings we suggest that what we observe is a link between maternal PTS and children's emotional processing, not a direct generation transfer of emotional processing abilities.

We have here demonstrated that traumatic events impact mother's mental health, which we suggest impacts her ability to scaffold, nurture and support her children [6] and leads to the observed effect, a reduced emotional processing ability of children. Poor emotional regulation, inappropriate emotional reactions and suboptimal parental strategies related to poor mental health might also contribute [8,10]. Together, these factors create a situation where children are at risk for developing lower levels of emotional processing than children that have mothers who are less effected by the war, or live in safe communities without experiences of war and refuge.

What can be done to buffer against the negative effects of poor maternal mental health on child development? One potential solution is to target mothers' PTSD symptoms. Effective treatments exist and have been demonstrated to alleviate symptoms in 60–80% of cases [36]. However, it has also been noted that refugees, often owing to co-morbidity, do not always reach such levels of treatment effects [37]. In Western contexts, several cost-effective treatments exist [38], but access to them are limited for Syrian refugees [39]. Even efforts like telepsychiatry, though sought after, especially by women, is limited owing to lack of internet security and an uncertainty about what it entails [39]. Cost effective, culturally sensitive group sessions lead by local stakeholders and community trainers might be an efficient and cost-effective way to alleviate PTSD symptoms in refugee populations [40]. Although preliminary work has demonstrated large effect sizes ($d = 1.93$ [41]), more work is needed with bigger samples before firm conclusions can be drawn.

On the other hand, making lives easier for mothers might be more effective, and have larger positive effects on the development of their children, than directly targeting PTSD symptoms. Supporting mothers financially, with work opportunities, or health insurance, might increase their quality of life and buffer against the observed negative effects [42]. It is also possible to support families and children with early access to schools [43], for communities to provide parental support programmes [7,44], and for children to be given space and time to play, explore and interact with others in a safe environment [45].

Before concluding, a series of caveats must be mentioned. First of all, when describing the effect of a reduction in mental health of mothers on child development we are referring to a change in a statistical model. As the current study is cross-sectional and report data from a single time-point we are not directly observing change in mothers mental health as a function of time. Future work, following families over time ideally in combination with active intervention programmes, need to be completed before we can conclude that a reduction in individual mothers PTSD symptoms is causally related to an increase in children's face processing.

Because a registry of refugees living in Turkish communities is not available, the current study was forced to rely on an opportunistic sampling procedure where each family helped in recruiting other participants. It would, of course, have been much better with a more rigorous sampling procedure in which a random, representative sample of families was invited to participate. Perhaps this is one reason why most studies focus on refugee camps, allowing more rigorous sampling procedures. Each approach has its merits, random samples of families in refugee camps offer a larger degree of generalization within this group, but makes it difficult to generalize to other settings where most refugees from Syria live. Investigating the research question from the current study in multiple settings is a natural next step.

It is important to note that the current study does not provide information about the mechanisms behind the observed generational transfer from maternal mental health to emotional processing in their children. We hope that future work will add clarity on this, ideally including, in addition to the measures reported above, behavioural observations or self-reports of parental practices.

In addition, direct information about children's own war experience is lacking. Future work would benefit from this information, as it would strengthen the analysis and increase the specificity of the

results. At the same time, doing so requires the ability to support the children over time in cases where this causes negative emotions/reactions, something that was not possible in the current study.

Finally, all questionnaires and instructions were translated to Arabic, and back-translated to English, by the research assistants working on the study. It would have been preferable to have standardized Arabic versions of all tests (including databases of emotional faces from the region), unfortunately this was not available, and represents another area of potential improvements for future studies.

In conclusion, we demonstrate, in a sample of 100 Syrian families living in Turkish communities, that high levels of maternal PTS negatively impact children's emotional processing development. At the same time, fathers' post-traumatic experiences did not impact child development in this sample. The results further suggest that a small improvement in the mental health of mothers might result in a positive improvement in the social cognitive development of their children. By providing targeted support to mothers it might be possible break the generational chain and help future generations grow to their full potential.

Ethics. The study was approved by the regional ethics review board in Sweden (2018-395) and the Necmettín Erbakan Universitesi in Turkey (2019/17). Written and verbal informed consent was obtained from all participants.

Data accessibility. The data used to run the above mentioned analysis is available at OSF (https://osf.io/qjrav/?view_only=2fd2c86db8614615b93b33fc899f8239) [30].

Authors' contributions. All authors participated in the design of the study and approved the final version of the manuscript. S.H. and J.H. implemented the test protocol, D.C.K. was in charge of the data collection in Turkey, M.L. and S.H. analysed the data. G.G. wrote the first version of the manuscript with critical input from all authors.

Competing interests. We declare we have no competing interests.

Funding. Funding for this project was provided by the Wallenberg Foundation (grant no. 2012.0120) awarded to G.G. S.P. was supported by the National Institute of Mental Health (grant no. MH61285) and the National Institute of Child Health and Human Development (grant no. U54 HD090256). J.H. was supported by the Swedish Research Council (grant no. 2015-06564).

Acknowledgements. We highly appreciate the work of Syrian research assistants Hassan Alali and Warda Bilal.

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
