## [Peer Review File · Royal Society Open Science]

Review History

RSOS-210362.R0 (Original submission)

Review form: Reviewer 1

Is the manuscript scientifically sound in its present form?

Yes

Are the interpretations and conclusions justified by the results?

Yes

Is the language acceptable?

Yes

Do you have any ethical concerns with this paper?

No

Have you any concerns about statistical analyses in this paper?

No

Recommendation?

Major revision is needed (please make suggestions in comments)

Comments to the Author(s)

I enjoyed reading this paper, which addresses a topic of utmost importance, not least in this day and age of blossoming migration. The study uses an impressive sample size and an intergenerational design. The key findings are that many of the parents studied have been heavily traumatized and most suffer from PTSD in the clinical realm, and that high maternal PTSD scores predict child emotional processing deficits. While the former findings are not surprising, it continues to be important to document this. The latter intergenerational findings showcase why it's important. In addition, the paper is well-written on the whole, and the analyses are appropriate as well as competently performed. Despite this overall positive assessment, I have several critical remarks/suggestions. Following these will undoubtedly strengthen the paper.

(1) The post-traumatic symptom scores and Harvard trauma scale scores are listed as dependent variables in the method section, but then (in the analysis section) stated as predictors of emotion processing. The former appears erroneous and should be adjusted.

(2) I can't see in the results section that the authors controlled for mothers' emotion processing when analyzing PTS effects on children's emotion processing, although they stated earlier in the paper that they would (i.e., to control for family social cognition). Also, despite not doing this, they take the null findings from fathers' post-traumatic symptom scores to indicate that what they have observed is not just some shared family social cognition. Why not include the pertinent parent's relevant emotional processing score as covariate instead?

(3) The authors state that this study is experimental. I suppose you could argue that randomization of the order in which the emotions are presented makes this a within-subject experiment. (It is clearly not a between-subject experiment, as there was no control group.) However, the key predictor/independent variables (parental trauma and PTS scores) were understandably never manipulated. Therefore, I read this as an intergenerational, correlational study, showing that natural variation in mothers' PTS scores predict natural variation in their children's emotional processing skills. These links have not been experimentally demonstrated so the authors should omit the experimental aspirations. The study is strong enough as is. False advertising only serves to weaken it. One implication of this is that the occasional slips into causal lingo (x impacts y) should be replaced with appropriate terminology (e.g., predicts, is linked or related to).

(4) Why did maternal but not paternal post-traumatic scores predict child processing? This question is not addressed head-on but should be. The first speculation that comes to my mind is that this reflects mothers being more actively engaged in interaction with their children (i.e., a gender-stereotyped caregiving arrangement).

(5) I found myself struggling how to best understand the effect sizes observed between maternal post-traumatic symptom scores and child outcomes. On the one hand, the CIs are comfortably remote from 0 (or above) and the reference to one drop in maternal scores being equivalent to one year of child development gives the impression of a substantive effect size. On the other, the odds ratios are pretty modest (e.g., .97 and 1.05). I assume many readers will be similarly confused so the authors should help readers by explicitly noting whether the effects observed are small, moderate, or large, according to available statistical conventions.

(6) Finally, the linking mechanism(s) remain unknown, indeed not studied here, which should be added as another caveat. The authors offer several viable suggestions (all relating to aspects of the caregiver's/mother's behavior) in the discussion and may want to encourage future research that addresses these by including pertinent caregiving measures (ideally behavioral ones).

(7) (There are some minor language glitches (e.g., "mothers that..." instead of "mothers who...").

Review form: Reviewer 2

Is the manuscript scientifically sound in its present form?

Yes

Are the interpretations and conclusions justified by the results?

Yes

Is the language acceptable?

Yes

Do you have any ethical concerns with this paper?

No

Have you any concerns about statistical analyses in this paper?

No

Recommendation?

Accept with minor revision (please list in comments)

Comments to the Author(s)

Comments for Authors

This study aimed to bring understanding of the relation between mothers' or fathers' experience of war and children's emotional processing. It was found that mothers', but not fathers', post-traumatic stress symptoms influence children's recognition of facial expressions. The topic has a great societal importance and has received only limited amount of research interest. This study could serve as a relevant opening for the research area. I have few questions and suggestions for the authors to consider when revising the manuscript for publication.

I would rethink whether the title is too general for using the concept of social cognition in the title as only one task i.e., recognition of facial expressions was used.

At the end of the introduction, there is no hypotheses. This could be a deliberate choice due limited number of previous studies, but could it be possible to assume something and if not state that explicitly?

In the methods, it is said that the parents helped children with the task if needed. I assume they only helped in understanding the instruction not in the recognition task per se. Could this be clarified? In addition, the instruction for the children was not entirely clear. Was the speeded response required or was the answering time unlimited? It could have also been informative for

the readers to get even a rough idea of the length of the family visits (e.g., a range of the length) although it varied a lot.

I was a bit puzzled how it was controlled whether the children's own war experience influenced the recognition of emotional expressions. Was that kind of information available? If not, I think that would be important issue to bring out in the discussion.

In my view, the discussion could focus a bit more on emotion processing instead of suggestion for intervention/treatment for mothers albeit that has a great societal impact. It would be interesting to have some discussion of findings of different facial expressions. For example, what do the authors think of the lack of significant findings when recognising anger? Could it be that these children have developed as somewhat oversensitive to recognise anger due their experiences? Was there something interesting when analysing the pattern of mistakes (e.g., biased towards negative expressions)? As a final thing, I wonder whether the lack of findings between adult's war experience and expression recognition related to the fact that the task might have not been sensitive enough for the adults.

Decision letter (RSOS-210362.R0)

Dear Professor Gredebäck

On behalf of the Editors, we are pleased to inform you that your Manuscript RSOS-210362 "Social cognition in refugee children: An experimental cross-sectional study with Syrian families in Turkish communities." has been accepted for publication in Royal Society Open Science subject to minor revision in accordance with the referees' reports. Please find the referees' comments along with any feedback from the Editors below my signature.

Please submit your revised manuscript and required files (see below) no later than 7 days from today's (ie 28-Jun-2021) date. Note: the ScholarOne system will 'lock' if submission of the revision is attempted 7 or more days after the deadline. If you do not think you will be able to meet this deadline please contact the editorial office immediately.

on behalf of Dr Teodora Gliga (Associate Editor) and Essi Viding (Subject Editor)
 openscience@royalsociety.org

Associate Editor Comments to Author (Dr Teodora Gliga):

I have now received comments from two reviewers. They both find great merit in your manuscript, especially in terms of its societal importance. They also request a number of clarifications and more in depth discussion of findings. I particularly urge you to 1. indicate clear hypothesis at the outset; 2. adjust your description of the importance of effects based on your effect sizes and 3. expand the discussion of findings with a focus on the putative mechanisms underlying the inter-generational effects (and their specificity to mother). I also agree that the title should reflect the narrower scope of the social abilities tested. These requests are not substantial so I do not anticipate having to return the manuscript to the reviewers.

Reviewer comments to Author:

Reviewer: 1

Comments to the Author(s)

I enjoyed reading this paper, which addresses a topic of utmost importance, not least in this day and age of blossoming migration. The study uses an impressive sample size and an intergenerational design. The key findings are that many of the parents studied have been heavily traumatized and most suffer from PTSD in the clinical realm, and that high maternal PTSD scores predict child emotional processing deficits. While the former findings are not surprising, it continues to be important to document this. The latter intergenerational findings showcase why it's important. In addition, the paper is well-written on the whole, and the analyses are appropriate as well as competently performed. Despite this overall positive assessment, I have several critical remarks/suggestions. Following these will undoubtedly strengthen the paper.

(1) The post-traumatic symptom scores and Harvard trauma scale scores are listed as dependent variables in the method section, but then (in the analysis section) stated as predictors of emotion processing. The former appears erroneous and should be adjusted.

(2) I can't see in the results section that the authors controlled for mothers' emotion processing when analyzing PTS effects on children's emotion processing, although they stated earlier in the paper that they would (i.e., to control for family social cognition). Also, despite not doing this, they take the null findings from fathers' post-traumatic symptom scores to indicate that what they have observed is not just some shared family social cognition. Why not include the pertinent parent's relevant emotional processing score as covariate instead?

(3) The authors state that this study is experimental. I suppose you could argue that randomization of the order in which the emotions are presented makes this a within-subject experiment. (It is clearly not a between-subject experiment, as there was no control group.). However, the key predictor/independent variables (parental trauma and PTS scores) were understandably never manipulated. Therefore, I read this as an intergenerational, correlational study, showing that natural variation in mothers' PTS scores predict natural variation in their children's emotional processing skills. These links have not been experimentally demonstrated so the authors should omit the experimental aspirations. The study is strong enough as is. False

advertising only serves to weaken it. One implication of this is that the occasional slips into causal lingo (x impacts y) should be replaced with appropriate terminology (e.g., predicts, is linked or related to).

(4) Why did maternal but not paternal post-traumatic scores predict child processing? This question is not addressed head-on but should be. The first speculation that comes to my mind is that this reflects mothers being more actively engaged in interaction with their children (i.e., a gender-stereotyped caregiving arrangement).

(5) I found myself struggling how to best understand the effect sizes observed between maternal post-traumatic symptom scores and child outcomes. On the one hand, the CIs are comfortably remote from 0 (or above) and the reference to one drop in maternal scores being equivalent to one year of child development gives the impression of a substantive effect size. On the other, the odds ratios are pretty modest (e.g., .97 and 1.05). I assume many readers will be similarly confused so the authors should help readers by explicitly noting whether the effects observed are small, moderate, or large, according to available statistical conventions.

(6) Finally, the linking mechanism(s) remain unknown, indeed not studied here, which should be added as another caveat. The authors offer several viable suggestions (all relating to aspects of the caregiver's/mother's behavior) in the discussion and may want to encourage future research that addresses these by including pertinent caregiving measures (ideally behavioral ones).

(7) (There are some minor language glitches (e.g., "mothers that..." instead of "mothers who...").

Reviewer: 2

Comments to the Author(s)

Comments for Authors

This study aimed to bring understanding of the relation between mothers' or fathers' experience of war and children's emotional processing. It was found that mothers', but not fathers', post-traumatic stress symptoms influence children's recognition of facial expressions. The topic has a great societal importance and has received only limited amount of research interest. This study could serve as a relevant opening for the research area. I have few questions and suggestions for the authors to consider when revising the manuscript for publication.

I would rethink whether the title is too general for using the concept of social cognition in the title as only one task i.e., recognition of facial expressions was used.

At the end of the introduction, there is no hypotheses. This could be a deliberate choice due limited number of previous studies, but could it be possible to assume something and if not state that explicitly?

In the methods, it is said that the parents helped children with the task if needed. I assume they only helped in understanding the instruction not in the recognition task per se. Could this be clarified? In addition, the instruction for the children was not entirely clear. Was the speeded response required or was the answering time unlimited? It could have also been informative for the readers to get even a rough idea of the length of the family visits (e.g., a range of the length) although it varied a lot.

I was a bit puzzled how it was controlled whether the children's own war experience influenced the recognition of emotional expressions. Was that kind of information available? If not, I think that would be important issue to bring out in the discussion.

In my view, the discussion could focus a bit more on emotion processing instead of suggestion for intervention/treatment for mothers albeit that has a great societal impact. It would be interesting to have some discussion of findings of different facial expressions. For example, what do the authors think of the lack of significant findings when recognising anger? Could it be that these children have developed as somewhat oversensitive to recognise anger due their experiences? Was there something interesting when analysing the pattern of mistakes (e.g., biased towards negative expressions)? As a final thing, I wonder whether the lack of findings between adult's war experience and expression recognition related to the fact that the task might have not been sensitive enough for the adults.

===PREPARING YOUR MANUSCRIPT===

===PREPARING YOUR REVISION IN SCHOLARONE===

Author's Response to Decision Letter for (RSOS-210362.R0)

See Appendix A.

Decision letter (RSOS-210362.R1)

Dear Professor Gredebäck,

I am pleased to inform you that your manuscript entitled "Social cognition in refugee children: An experimental cross-sectional study of emotional processing with Syrian families in Turkish communities." is now accepted for publication in Royal Society Open Science.

on behalf of Dr Teodora Gliga (Associate Editor) and Essi Viding (Subject Editor)
openscience@royalsociety.org

Appendix A

Dear Teadora,

Thanks for the constructive comments provided by you and the two reviewers. We believe the manuscript will be much improved by the changes made. Below our response to each comment is marked with RESPONSE:

Best regards,

Gustaf (on behalf of all co-authors)

Associate Editor Comments to Author (Dr Teodora Gliga):

I have now received comments from two reviewers. They both find great merit in your manuscript, especially in terms of its societal importance. They also request a number of clarifications and more in depth discussion of findings. I particularly urge you to 1. indicate clear hypothesis at the outset; 2. adjust your description of the importance of effects based on your effect sizes and 3. expand the discussion of findings with a focus on the putative mechanisms underlying the inter-generational effects (and their specificity to mother). I also agree that the title should reflect the narrower scope of the social abilities tested. These requests are not substantial so I do not anticipate having to return the manuscript to the reviewers.

RESPONSE: 1) The hypothesis is now added towards the end of the intro. 2) A more detailed discussion based on effect size is now added to the discussion and results. 3) The discussion about putative mechanisms underlying the inter-generational effects between mother and child is now included in the discussion.

Reviewer comments to Author:

Reviewer: 1

Comments to the Author(s)

I enjoyed reading this paper, which addresses a topic of utmost importance, not least in this day and age of blossoming migration. The study uses an impressive sample size and an intergenerational design. The key findings are that many of the parents studied have been heavily traumatized and most suffer from PTSD in the clinical realm, and that high maternal PTSD scores predict child emotional processing deficits. While the former findings are not surprising, it continues to be important to document this. The latter intergenerational findings showcase why it's important. In addition, the paper is well-written on the whole, and the analyses are appropriate as well as competently performed. Despite this overall positive assessment, I have several critical remarks/suggestions. Following these will undoubtedly strengthen the paper.

(1) The post-traumatic symptom scores and Harvard trauma scale scores are listed as dependent variables in the method section, but then (in the analysis section) stated as predictors of emotion processing. The former appears erroneous and should be adjusted.

RESPONSE: Corrected.

(2) I can't see in the results section that the authors controlled for mothers' emotion processing when analyzing PTS effects on children's emotion processing, although they stated earlier in the paper that they would (i.e., to control for family social cognition). Also, despite not doing this, they take the null findings from fathers' post-traumatic symptom scores to indicate that what they have observed is not just some shared family social cognition. Why not include the pertinent parent's relevant emotional processing score as covariate instead?

RESPONSE: The first child analysis includes both fathers and mothers PTS, but as only mothers' mental health had a unique contribution this is the only factor that is kept for the second analysis.

(3) The authors state that this study is experimental. I suppose you could argue that randomization of the order in which the emotions are presented makes this a within-subject experiment. (It is clearly not a between-subject experiment, as there was no control group.). However, the key predictor/independent variables (parental trauma and PTS scores) were understandably never manipulated. Therefore, I read this as an intergenerational, correlational study, showing that natural variation in mothers' PTS scores predict natural variation in their children's emotional processing skills. These links have not been experimentally demonstrated so the authors should omit the experimental aspirations. The study is strong enough as is. False advertising only serves to weaken it. One implication of this is that the occasional slips into causal lingo (x impacts y) should be replaced with appropriate terminology (e.g., predicts, is linked or related to).

RESPONSE: We intended to use the term experimental to denote the use of an experimental task, not to imply that the study in its entirety is experimental. As most prior work is based on interviews and questionnaires this is an important point. However, we see that this can lead to confusion and have tried to clarify our terminology in the intro.

(4) Why did maternal but not paternal post-traumatic scores predict child processing? This question is not addressed head-on but should be. The first speculation that comes to my mind is that this reflects mothers being more actively engaged in interaction with their children (i.e., a gender-stereotyped caregiving arrangement).

RESPONSE: This is discussed briefly in the discussion, but without more information it is difficult to be too lengthy about this issue. Follow up studies are looking into this issue. But a new section on potential reasons for the maternal effect is now added, perhaps this gets us some way towards what the reviewer is asking.

(5) I found myself struggling how to best understand the effect sizes observed between maternal post-traumatic symptom scores and child outcomes. On the one hand, the CIs are comfortably remote from 0 (or above) and the reference to one drop in maternal scores being equivalent to one year of child development gives the impression of a substantive effect size. On the other, the odds ratios are pretty modest (e.g., .97 and 1.05). I assume many readers will be similarly confused so the authors should help readers by

explicitly noting whether the effects observed are small, moderate, or large, according to available statistical conventions.

RESPONSE: Thank you for this comment. Our intention with the comparison between the two effect sizes was to give the reader a sense of what an odds ratio of, for example, .97 for the effect of maternal PTS on child emotion processing was comparable with. That is, to anchor it against something a bit more intuitive, i.e. how emotion processing develops during one year of childhood. We realize that by doing so we might have unintentionally given the impression that this is a particularly large effect. The reviewer is indeed correct in noticing that these odds ratios are small in statistical terms. In the revised manuscript, we now emphasize the size of the effect both in statistical terms and as a comparison between development and maternal PTS in an effort to be more transparent.

(6) Finally, the linking mechanism(s) remain unknown, indeed not studied here, which should be added as another caveat. The authors offer several viable suggestions (all relating to aspects of the caregiver's/mother's behavior) in the discussion and may want to encourage future research that addresses these by including pertinent caregiving measures (ideally behavioral ones).

RESPONSE: Thanks for this point, it is now added as an additional caveat.

(7) (There are some minor language glitches (e.g., "mothers that..." instead of "mothers who...").

RESPONSE: Thanks, corrected.

Reviewer: 2

Comments to the Author(s)

Comments for Authors

This study aimed to bring understanding of the relation between mothers' or fathers' experience of war and children's emotional processing. It was found that mothers', but not fathers', post-traumatic stress symptoms influence children's recognition of facial expressions. The topic has a great societal importance and has received only limited amount of research interest. This study could serve as a relevant opening for the research area. I have few questions and suggestions for the authors to consider when revising the manuscript for publication.

I would rethink whether the title is too general for using the concept of social cognition in the title as only one task i.e., recognition of facial expressions was used.

RESPONSE: Thanks for this, a clarification has been added to the title.

At the end of the introduction, there is no hypotheses. This could be a deliberate choice due limited number of previous studies, but could it be possible to assume something and if not state that explicitly?

RESPONSE: The hypothesis is not more clearly spelled out towards the end of the intro.

In the methods, it is said that the parents helped children with the task if needed. I assume they only helped in understanding the instruction not in the recognition task per se. Could this be clarified? In addition, the instruction for the children was not entirely clear. Was the speeded response required or was the answering time unlimited? It could have also been informative for the readers to get even a rough idea of the length of the family visits (e.g., a range of the length) although it varied a lot.

RESPONSE: It is now clarified that parents sometimes helped with instructions. The duration of the family visits was not recorded, unfortunately.

I was a bit puzzled how it was controlled whether the children's own war experience influenced the recognition of emotional expressions. Was that kind of information available? If not, I think that would be important issue to bring out in the discussion.

RESPONSE: We have added this to the discussion, an important point.

In my view, the discussion could focus a bit more on emotion processing instead of suggestion for intervention/treatment for mothers albeit that has a great societal impact. It would be interesting to have some discussion of findings of different facial expressions. For example, what do the authors think of the lack of significant findings when recognising anger? Could it be that these children have developed as somewhat oversensitive to recognise anger due their experiences? Was there something interesting when analysing the pattern of mistakes (e.g., biased towards negative expressions)? As a final thing, I wonder whether the lack of findings between adult's war experience and expression recognition related to the fact that the task might have not been sensitive enough for the adults.

RESPONSE: We observe similar responses for all emotions but only some pass the classical 0.05 threshold. We would prefer not to make too much of the observed differences and only note that the effect is broad and not specific to a single emotion, as some papers and theories of emotional processing would suggest. There are also always more ways to analyze data but we would prefer not to go outside our initial analysis plan, if possible. The data is freely available if others would like to analyze the data in other ways.